# Alleviation of Tris(2-chloroethyl) Phosphate Toxicity on the Marine Rotifer *Brachionus plicatilis* by Polystyrene Microplastics: Features and Molecular Evidence

**DOI:** 10.3390/ijms23094934

**Published:** 2022-04-29

**Authors:** Wenqian Ma, Zijie Sun, Xin Zhang, Xuexi Tang, Xinxin Zhang

**Affiliations:** 1College of Marine Life Sciences, Ocean University of China, Qingdao 266003, China; 17853729216@163.com (W.M.); aszjyt@163.com (Z.S.); zxouc1@163.com (X.Z.); 2Laboratory for Marine Ecology and Environmental Science, Pilot National Laboratory for Marine Science and Technology, Qingdao 266237, China

**Keywords:** combined effect, microplastics, TCEP, *Brachionus plicatilis*, multi-xenobiotic resistance mechanism, transcriptomic, endogenous response

## Abstract

As emerging pollutants, microplastics (MPs) and organophosphorus esters (OPEs) coexist in the aquatic environment, posing a potential threat to organisms. Although toxicological studies have been conducted individually, the effects of combined exposure are unknown since MPs can interact with OPEs acting as carriers. In this study, we assessed the response of marine rotifer, *Brachionus plicatilis*, to co-exposure to polystyrene MPs and tris(2-chloroethyl) phosphate (TCEP) at different concentrations, including population growth, oxidative status, and transcriptomics. Results indicated that 0.1 μm and 1 μm MPs were accumulated in the digestive system, and, even at up to 2000 μg/L, they did not exert obvious damage to the stomach morphology, survival, and reproduction of *B. plicatilis*. The presence of 1 μm MPs reversed the low population growth rate and high oxidative stress induced by TCEP to the normal level. Some genes involved in metabolic detoxification and stress response were upregulated, such as ABC and *Hsp*. Subsequent validation showed that P-glycoprotein efflux ability was activated by combined exposure, indicating its important role in the reversal of population growth inhibition. Such results challenge the common perception that MPs aggravate the toxicity of coexisting pollutants and elucidate the molecular mechanism of the limited toxic effects induced by MPs and TCEP.

## 1. Introduction

Plastic pollution has become a global concern. In recent years, plastic production has continued to expand due to its characteristics of corrosion resistance, water resistance, and light weight [1], and the demand for plastic products has increased rapidly [2]. It is estimated that total plastic production will reach 1.1 billion tons in 2050 if plastic products continue to be widely used without effective control measures [3,4]. Plastic fragments in the aquatic environment are stable and difficult to collect and remove but can be degraded into small fragments [5]. Microplastics (MPs) with particle sizes smaller than 5 mm are the largest outcome of plastic pollution and are ubiquitous in the marine environment as emerging pollutants [6,7]. The abundance of MPs in surface sediments of the South Yellow Sea of China was as high as 4205 numbers/kg dry weight [8]. MPs are so mobile in water that they can be detected in the Antarctic and Arctic [9,10]. Due to their small size, wide distribution, and high bioavailability, MPs and nanoplastics (NPs, with a diameter of less than 100 μm) are gradually ingested by a variety of marine animals and enter marine animals at different trophic levels along the food chain; they have been detected in the digestive tracts and tissues of plankton [11], benthic organisms [12], fish [13], and marine bivalves [14]. It was reported that MP intake changed the body size and biomass of the marine ciliate *Uronema marinum* without particle size-selective effects [15], accumulated in the rotifer gut, and altered swimming behavior and reproduction [16,17]. MPs also increased satiety in marine large fish, reduced food consumption [18], and caused an imbalance in intestinal microflora. Moreover, exposure to polystyrene MPs for 7 days at 1000 μg/L induced neurotoxicity, oxidative stress, immune system activation, and metabolic disorders in juvenile zebrafish [19]. However, some studies have also reported that 1–6 μm polyethylene MPs have no acute toxicity toward *Brachionus plicatilis* and *Tigriopus fulvus* at up to 10 mg/L [20], while the growth of the alga *Raphidocelis subcapitata* was found to be promoted by polyethylene MPs at 130 mg/L [21]. These inconsistent results suggest that the effect of MPs depends on the texture, size, and concentration of plastic, as well as the organism species. In particular, some exorbitant concentrations completely divorced from reality were used. Thus, continued attention should be given to the effects of MPs at environmental concentrations on marine life, which will promote a comprehensive understanding of the marine ecological risks of MP and effective management of plastic use.

MPs and organic pollutants are likely to coexist in the aquatic environment [22]. Due to their negatively charged surfaces and high affinity for organic pollutants, MPs can interact with organic pollutants as carriers, resulting in a higher concentration of organic pollutants on MPs than in seawater [23]. It has been speculated that the negative effects of the absorbed organic pollutants on organisms might be deteriorated by MPs. For instance, polychlorinated biphenyls (PCBs), polycyclic aromatic hydrocarbons (PAHs), and discarded pharmaceuticals [24] can easily adsorb onto the surface of MPs, resulting in an increase in bioavailability [25]. It was found that MPs enhanced the toxicity of phenanthrene and 9-nitroanthracene toward zebrafish [26,27]. The toxicity of benzo[a]pyrene on the early growth and development of marine medaka embryos and larvae was also enhanced in the presence of MPs [28]. However, a study by Beiras [29] showed that sea urchin larvae actively ingested MPs but did not increase the toxicity of 4-*n*-nonylphenol even at concentrations much higher than those in the marine environment. Co-exposure to MPs and pyrene had little effect on the feeding and swimming performance of juvenile barramundi (*Lates calcarifer*) [30]. More strikingly, Zhang et al. [31] found that MPs mitigated the developmental toxicity of triphenyl phosphate (TPhP) to marine medaka (*Oryzias melastigma*) larvae and restored their locomotor activity. Additionally, MPs can reduce the reproductive toxicity and metabolic damage of tributyltin to the rotifer *Brachionus koreanus* under suitable nutritional conditions [32]. Since the effect of MPs on the toxicity of organic pollutants varies widely, additional studies are needed to reveal their combined effects and further evaluate the environmental risks of MPs pollution to marine ecosystems.

In recent years, as a replacement for brominated flame retardants, organophosphorus esters (OPEs) with excellent flame-retardant properties and plasticity have been primarily utilized in various kinds of products, including plastic products, textile coatings, electronic equipment, paints, and lubricants [33,34]. OPEs are physically bound to these polymeric materials; thus, they are easily released into the environment through volatilization, abrasion, or dissolution [35], and they have become emerging contaminants in various environments [36]. As a typical chlorinated OPE, tris(2-chloroethyl) phosphate (TCEP) is often detected in aquatic environments because of its water solubility [37]. It was reported that TCEP was the most abundant OPEs in the coastal waters of the Pearl River Estuaries in China [35], and the concentration was as high as 87.4 μg/L in the effluents of wastewater treatment plants in Japan [38]. TCEP can be transferred through aquatic food chains and cause persistent harm to higher-trophic-level organisms [39], including reproductive toxicity, developmental toxicity, neurotoxicity, mutagenicity, and carcinogenicity [40,41]. For example, Hu et al. [41] reported that TCEP can cause a reduction in body length, endocrine disruption of the thyroid gland, and delayed hatching of zebrafish. TCEP also reduced cell viability and altered gene expression in the protozoan *Tetrahymena thermophila* [42]. However, limited information is known about the combined effect of MPs and TCEP on marine organisms.

As a marine rotifer, *Brachionus plicatilis* has been used as a test species since it has a small size, simple structure, short life cycle, ease of cultivation, high genetic homogeneity, and high fertility, as well as its ecological significance [43]. Some reports have shown that *B. plicatilis* has high sensitivity to pollutants [44,45]. This is because *B. plicatilis* likely possesses a unique mode of defense mechanism against exogenous substances compared to other aquatic species [46]. In the present study, we first evaluated the effects of MPs of two different sizes (0.1 μm, 1 μm) at environmental concentrations on key life-cycle parameters of *B. plicatilis*. Next, the combined effect of the 1-μm MPs and TCEP on rotifers was estimated, and the intrinsic mechanisms were explored on the basis of changes in the transcriptome and subsequent validation. The results will promote a better understanding of the effects of MPs at real concentrations on marine organisms and provide insights into the joint risk assessment of MPs and organic pollutants in the marine environment.

## 2. Results

### 2.1. Ingestion of MPs in Rotifers

The occurrence of fluorescently labeled MPs in rotifers after 24 h of exposure to different concentrations of 0.1-μm MPs and 1-μm MPs (20, 200, and 2000 μg/L) is displayed in Figure 1. For both sizes of MPs, the MPs emerged in the digestive system of rotifers relative to the control, such as the mastax, stomach, and intestines, even at an environmental concentration of 20 μg/L. Moreover, the fluorescence intensity obviously increased as the MP concentration rose (Appendix A), further suggesting that more MP particles entered the rotifers as the MP concentration increased. The statistical results of fluorescence intensity in rotifers among different groups are shown in Appendix A. 

### 2.2. Stomach Microstructural Observations

In the controls, a rotational ellipsoid-typed stomach was observed (Figure 2A). The stomach parietal cell was plump and arranged loosely, and the cilia inside the stomach were apparent. In the 0.1-μm MP (2000 μg/L) group, the stomach became more narrow, but the stomach parietal cell, cilia inside the stomach, and stomach lumen were not altered compared to control (Figure 2B). According to Figure 2C, no significant changes were observed in stomach microstructure between the 1-μm MP and control groups. It appeared that MPs of both sizes at 2000 μg/L caused no obvious damage of stomach morphology.

### 2.3. Changes in the Life History Traits of Rotifers in Response to Two Sizes of MPs

The age-specific survivorship of rotifers in response to 0.1-μm MPs is shown in Figure 3A. It is obvious that the majority of rotifers survived for the first 7 days and subsequently died over time. No difference occurred between the control and 0.1-μm MP treatments, even though the concentration of MPs was up to 2000 μg/L (*df* = 3, *F* = 0.017, *p* = 0.864, Figure 3B). Accordingly, lifespan was not changed by all concentrations of 0.1-μm MPs. As displayed in Figure 3C,D, no obvious changes in daily offspring production and the number of total offspring were observed between the control and 0.1-μm MP groups at any test concentration. According to Figure 3E, the survival rate of rotifers in the group of 1-μm MPs at 200 μg/L was lower than that of the control. All rotifers were dead by 15 days. The lifespan of rotifers was also reduced by 1-μm MPs at 200 μg/L (*df* = 3, *F* = 6.644, *p* < 0.05, Figure 3F). The daily production of offspring and the number of total offspring were not changed by the 1-μm MPs at any test concentration (Figure 3G,H). The statistical results are summarized in Appendix A.

### 2.4. The Combined Effect of 1-μm MPs and TCEP on the Population Growth and Oxidative Status of Rotifers

The population growth curve of rotifers in response to co-exposure to 1-μm MPs and TCEP at environmentally relevant concentrations is shown in Figure 4A, and the parameters after fitting to the logistic model are shown in Figure 4B,C. Rotifers entered the exponential period at 3 days, and the growth rate slowed down on the 11th day and 12th day for all experimental groups when the Tp value was not altered (Figure 4B). Compared to controls, the K value significantly decreased in the 100 μg/L TCEP group but did not change in the 1-μm MP and 1-μm MP + TCEP groups at environmental concentrations (Figure 4C). For the groups treated with 1-μm MPs and TCEP at high concentrations, no significant changes were observed in the population density of rotifers among all experimental groups during the first 7 days (Figure 4D). Later, 65 mg/L TCEP reduced the number of individuals relative to the control. For Tp (d), single exposure to 65 mg/L TCEP or 1-μm MPs did not have a negative effect, but an extension was observed in the co-exposure group (*df* = 3, *F* = 2.773, *p* < 0.05, Figure 4E). The K value of the rotifer population was significantly inhibited by 65 mg/L TCEP (*df* = 3, *F* = 6.361, *p* < 0.05), while it did not change in the 1-μm MP group or the 1-μm MP + TCEP group relative to the control (Figure 4F). After 24 h of exposure, a significant increase in DCF fluorescence was found in the 1-μm MP, TCEP, and 1-μm MP + TCEP groups compared to the control (Figure 5A). The MDA level in rotifers was elevated by TCEP but unaffected by 1-μm MPs and 1-μm MP + TCEP co-exposure (Figure 5B). The statistical results of population growth and the oxidative status of rotifers among different groups are shown in Appendix A and Appendix A, respectively.

### 2.5. Transcriptomics Response

In this study, 45.55–51.31 M bases of raw reads were obtained from the exposed groups and controls. A total of 45.08–50.62 M clean bases were obtained after removing low-quality reads. The Q30 and GC values are shown in Appendix A. A total of 24,269 unigenes that were clustered had an N50 of 1963 bp and an average length of 1319.13 (Appendix A). The sequence length was mainly distributed in the range larger than 2000 bp, followed by 0–500 bp (Appendix A). The numbers of unigene annotations in seven databases, NR, Swiss-Port, KEGG, KOG, eggNOG, GO, and Pfam, were 12,202, 10,094, 4121, 9111, 10,900, 9323, and 10,565, respectively (Appendix A). The percentage similarity of unigenes with respect to *Lingula anatine* (7.49%), *Hydra vulgaris* (3.96%), *Crassostrea virginica* (3.93%), and *Crassostrea gigas* (3.28%) (Appendix A) indicated both the abundant genomic information and the phylogenetic relationships for those species.

A Venn diagram of the transcriptome is displayed in Appendix A. Compared to the control, there were 1166, 595, and 757 DEGs identified by transcriptomic analysis in the 1-μm MP, TCEP, and 1-μm MP + TCEP groups, respectively, including 652, 316, and 382 downregulated genes, respectively. A total of 22.64% and 30.08% of DEGs in the 1-μm MP group and TCEP group, respectively, were shared with those of the combined exposure (1-μm MP + TCEP groups), suggesting an additive gene response to 1-μm MPs and TCEP. According to Appendix A, the qPCR results of some DEGs were well matched with the RNA-Seq results, suggesting their reliability.

The results of GO enrichment showed that 912, 225, and 557 GO terms (*p* < 0.05) were significantly enriched by exposure to 1-μm MPs, TCEP, and 1-μm MPs + TCEP, respectively. For the 1-μm MP group, the majority of GO terms were biological processes “caveola assembly”, cellular component “rough extracellular space” and “polysomal ribosome” and molecular function “glutathione transferase activity” (Figure 6A). For TCEP exposure, the majority of GO terms were biological process “protein localization to plasma membrane” and molecular function “cadherin binding” (Figure 6B). For 1 μm MP + TCEP exposure, the majority of GO terms were biological process “cholesterol efflux”, cellular components “rough endoplasmic reticulum” and “polysomal ribosome”, and molecular functions “glutathione transferase activity” and “peptide binding” (Figure 6C). This result suggested a difference in the transcriptomic responses of rotifers after single and combined exposure.

The KEGG pathway analysis showed that “phagosome” (*p*-value: 1.25 × 10^−^^4^) and “NOD-like receptor signaling pathway” (*p*-value: 1.93 × 10^−4^) were the most significant pathways in the 1-μm MP group, possibly related to the toxicity pathway (Appendix A). The pathways were mainly mapped to “oxidative phosphorylation” (*p*-value: 1.43 × 10^−4^) in the TCEP group (Appendix A). For 1-μm MP + TCEP exposure, the most significant pathway was “sulfur metabolism” (*p*-value: 4.06 × 10^−3^). For the 1-μm MP group and 1-μm MP + TCEP group, the pathway of metabolism of xenobiotics by cytochrome P450 was shared (Appendix A).

### 2.6. Changes in Multi-Xenobiotic Resistance System Activities

Images of rotifers after staining with rhodamine B and calcein AM in vivo are shown in Figure 7A–D. No significant differences were found for rhodamine B retention in the 1-μm MPs group and TCEP group after 24 h of exposure compared to the controls, but a declining tendency occurred in the 1-μm MP + TCEP group (Figure 7E). This low fluorescence value for cellular retention implied a high efflux activity of P-glycoprotein (P-gp) induced by the co-exposure of two pollutants. Changes in calcein AM fluorescence per mg of protein in rotifers after exposure to 1-μm MPs, 1-μm MPs + TCEP, and TCEP are displayed in Figure 7F. Compared with the control group, calcein AM fluorescence decreased significantly in all treatments. Thus, multidrug resistance-associated protein (MRP) activities were stimulated to transport these xenobiotics from inside the cell to outside. All three exposure groups stimulated the activities of glutathione *S*-transferase (GST) relative to the control (*df* = 3, *F* = 31.394, *p* < 0.05). However, for both single 1-μm MP exposure and 1-μm MP + TCEP co-exposure, activities were higher than for TCEP exposure. In the co-exposure group, the activity of this enzyme increased 1.66-fold compared to the control (Figure 7G). The statistical results of MXR activity of rotifers among different groups are shown in Appendix A.

## 3. Discussion

MPs are ubiquitous in the marine environment, which has raised extensive concern about their ecological effects on marine organisms. Although many studies have been performed, there are still few data about the effect of MPs at environmental concentrations on marine life. MPs of two sizes were used in this experiment because of their worldwide distribution [47]. Additionally, it has been reported that *B. plicatilis* can ingest food particles with sizes less than 20 μm [48]; thus, 0.1-μm and 1-μm MPs can be easily ingested by rotifers without selective grazing. Fluorescently labeled MPs particles were mainly concentrated in the stomach and mastax, followed by the intestines. The stomach, which presents a form of syncytium and is lined with tiny cilia, plays an important role in food digestion and uptake by rotifers. MP accumulation in the stomach exerted no structural damage, as revealed by paraffin sections of rotifer stomach samples after exposure to MPs at 2000 μg/L. In this study, no obvious changes were found in the reproduction, survival, and population growth of rotifers for both sizes of MPs in the range of 20–2000 μg/L. This result was contrary to the marine copepod *Tigriopus japonicus* [49] but in accordance with *Daphnia magna* [21,50], suggesting a species-specific response of zooplankton to MPs. It was speculated that the long-lasting consequence was a disturbance of the zooplankton community. 

Although 1-μm MPs at 2000 μg/L did not change the lifespan, total produced offspring number, or population growth of rotifers, transcriptomic analysis identified 1166 DEGs. A total of 21 DEGs could be matched in *Brachionus*, and most of them were upregulated, including several GSTs and hatching enzyme-like proteins. Notably, the most identified GO term for molecular function was “glutathione transferase activity”. It has been reported that *B. plicatilis* has 24 GSTs that can be divided into the GST subfamilies alpha, omega, sigma, and zeta [51]. In the present study, the expression of five GSTs was upregulated by 1-μm MPs at 2000 μg/L, e.g., GST S4 (TRINITY_DN7992_c0_g1_i1_1) and GST S7 (TRINITY_DN4306_c0_g1_i1_1). In accordance with the transcriptional changes, GST activities also significantly increased. GSTs are important members of xenobiotic detoxification enzymes, in the presence of which water-soluble conjugated compounds are generated by catalyzing the conjugation of glutathione (GSH) to these electrophilic compounds through thioether linkages, which are more easily excreted from the organism [52,53]. It has been reported that GST can also react with excess ROS to relieve oxidative stress [54,55]. The consequent development of this protection was to minimize the intracellular stress [56]. Jeong et al. [57] also found that a significant increase in GST activity was observed in the rotifer *B. koreanus* in response to polystyrene microbeads, with a negative correlation between the size and GST activity (i.e., smaller size induced higher GST activity). Additionally, 1-μm MP groups exhibited upregulated expression levels of two genes, *Hsp*70 (TRINITY_DN14531_c0_g1_i1_2) and *Hsp90* alpha1 (TRINITY_DN8816_c0_g2_i1_2), which encode heat-shock proteins. In particular, *Hsp70* participates in a wide range of cellular functions, including protein folding and control of regulatory proteins [58,59], and its expression level can be regulated by a series of environmental stressors, such as heat and metal [60]. This molecular evidence may explain why the survival and reproduction of rotifers were not affected by 2000 μg/L MPs. 

TCEP at 100 μg/L (environmental concentration) and 65 mg/L was found to inhibit the population growth of rotifers, as evidenced by a decline in the K value. These results are in accordance with Hao et al. [42], who found that exposure to 0.044 mg/L TCEP for 5 days decreased the theoretical population of the aquatic protozoan *Tetrahymena thermophila*, which was attributed to proteasome dysfunction, thereby disrupting the cell cycle and further inhibiting the proliferation of cells. Surprisingly, the co-exposure of MPs and TCEP did not alter the K value, suggesting that the inhibition of population growth induced by TCEP was alleviated. Accordingly, this study found that an increase in ROS levels occurred in all treatments compared to the blank control; however, oxidative injury only occurred in the TCEP group. Previous studies have found that MPs might enhance the toxicity of other pollutants in marine organisms by increasing their bioavailability and bioaccumulation [61,62]. However, in this study, TCEP toxicity toward the population growth and oxidative status of rotifers was reversed to the normal level with controls in the presence of 1-μm MPs. Pollutants can enter rotifers through the corona, in which cilia rotate to form vortices and cross the cuticular surface by diffusion, subsequently bioaccumulating in the body and exerting a negative influence [63,64]. TCEP has been found to adhere to plastic polymers, with sorption increasing over time and approaching equilibrium [65]. Due to their large size, the agglomerates of 1-μm MPs and TCEP could not pass across the cuticular surface of rotifers at an appreciable rate. Thus, the low bioavailability of free TCEP might be responsible for the reduced toxicity observed in this study.

A total of 595 and 757 DEGs were identified by transcriptomic analysis in the TCEP and 1-μm MP + TCEP groups, respectively. There was an uncoupling between the number of DEGs and the degree of injury. This phenomenon was also observed in a study of the Pacific white shrimp *Litopenaeus vannamei* exposed to imidacloprid [66]. Meanwhile, a metabolic network was generated from some DEGs that could be matched *Brachionus* to clearly illustrate these changes from an overall perspective (Figure 8). MutS Homolog 5 (*MSH5*) is involved in DNA mismatch repair and meiotic recombination, and, in this study, one homologous gene *MSH5* (TRINITY_DN2622_c0_g1_i1_1) was downregulated by TCEP exposure and 1-μm MP + TCEP co-exposure. However, 1-μm MP + TCEP co-exposure upregulated the expression of another *MSH5* (TRINITY_DN4098_c0_g1_i1_2). As a cellular defense mechanism, the multi-xenobiotic resistance mechanism (MXR) is considered the first line of defense against endogenous and exogenous toxic substances in aquatic organisms. P-gp and MRP are the main components in this system [67]. P-gp and MRP belong to the family of ATP-binding cassette (ABC) transmembrane transport proteins that actively expel heterologous substances using energy by hydrolyzing ATP [68,69]. Seventy-three ABCs were found for *B. plicatilis* [70]. Co-exposure to 1-μm MPs and TCEP upregulated the expression of three ABC genes, while ABC genes were upregulated in the TCEP group. *Abcb1*, which encodes P-gp, was significantly expressed among all treatments. Similar regulation was also found in copepods (*T. japonicus*) exposed to cadmium, copper, and zinc [71] and in sea urchins (*Strongylocentrotus purpuratus*) exposed to inorganic mercury [72]. In the 1-μm MP + TCEP group, an increase in this efflux pump activity was also found, suggesting an important role of P-gp in rotifers in response to combined pollution. Thus, more TCEP could be pumped out of the cell. Moreover, compared to TCEP exposure alone, 1-μm MP + TCEP stimulated the expression of *Hsp70* (TRINITY_DN14531_c0_g1_i1_2) and *Hsp90* alpha1 (TRINITY_DN8816_c0_g2_i1_2). Overall, the coherence of multiple protection pathways determined different responses to 1-μm MP, TCEP, and 1-μm MP + TCEP stress, and P-gp activities and transcriptional changes in *HSPs* and *MSH5* might be responsible for the limited joint effect of MPs and TCEP.

## 4. Materials and Methods

### 4.1. Cultivation of B. plicatilis

Resting eggs of *B. plicatilis* (a strain of East China Sea) were incubated in Erlenmeyer flasks with sterile seawater in an illuminated incubator (PGX-330A-22H, Ningbo LIFE Instrument, Ningbo, China). The incubation conditions were kept at 25 ± 1 °C under a 12 h/12 h light/dark photoperiod with a PAR intensity of 60 μmol photons/(m^2^·s). After hatching, rotifers were cultured in flasks with sterile seawater that contained *Chlorella vulgaris* Beij as a live diet (approximately 1 × 10^6^ cells/mL) and changed every 3 days. Prior to the experiment, rotifers at a density of 10 individuals/mL were acclimated for half a month, and only robust rotifers with amictic eggs were selected later for normal experiments. The experimental system was based on the protocol of American Society for Testing and Materials (ASTM) named “Standard Guide for Acute Toxicity Test with Rotifer *Brachionus* (Designation: E1440-91)” [73].

### 4.2. Chemicals

Two different sizes of polystyrene microspheres (0.1 and 1 μm, water 1:1 emulsion, 2.5% (*w*/*v*)) were purchased from Tianjin DaE Technology Co., Ltd. (Tianjin, China). The morphology of these two particles photographed by electron microscopy is shown in Appendix A. The images indicate that MPs were virgin, monodisperse, and solid at the size of 0.1 μm and 1 μm. Prior to use, the solution was shaken firstly and then blended for 30 s to ensure the particles were evenly distributed. For intake tests, green fluorescent-labeled polystyrene microspheres with a ratio of water and emulsion of 1:1 were used; the excitation wavelength was 488 nm, and the emission wavelength was 518 nm. TCEP solution (CAS number: 115-96-8, analytical grade, purity ≥ 97%) was obtained from Sigma-Aldrich (Saint Louis, MO, USA). Dimethyl sulfoxide (DMSO, GC grade, purity ≥ 99.0%, liquid, Sigma-Aldrich, Saint Louis, MO, USA) was chosen as the cosolvent for TCEP, which was stored in the dark at room temperature. The maximum no-effect concentration of DMSO on rotifers was 0.8% (*v*/*v*), and the concentration of DMSO used throughout the experiment was less than 0.2% (*v*/*v*).

### 4.3. Impact of Different Sized MPs on Key Life History Parameters of B. plicatilis

#### 4.3.1. The Ingestion Test for MPs of Two Sizes

The concentrations of MPs at two sizes were 20, 200, and 2000 μg/L in the experiment. The first concentration (20 μg/L) represents the present level in the marine environment [31], while the other two concentrations represent high concentrations in the future without effective control of plastic usage. The corresponding numbers of particles per mL for the three concentrations were as follows: (1) 0.1 μm: 1.40 × 10^7^, 1.40 × 10^8^, and 1.40 × 10^9^ items/mL; (2) 1 μm: 1.40 × 10^4^, 1.40 × 10^5^, and 1.40 × 10^6^ items/mL. The group with only sterile seawater was considered a control. Active and robust rotifer larvae were selected for the MP exposure experiment. One milliliter of test solution and 10 rotifers were added into one well of a 24-well plate, and each group was analyzed in triplicate. The culture conditions were similar to those described in the hatching process. After exposure for 24 h, the rotifers were collected and washed with clean sterilized seawater to remove residual fluorescent microbeads on the surface. Then, the samples were fixed with 4% glutaraldehyde and photographed under a fluorescence microscope (Olympus IX 71, Tokyo, Japan).

#### 4.3.2. Microstructural Analysis of Rotifer Stomach

The stomach is the most important part in the digestive system of *B. plicatilis*, and its structure was observed by paraffin sections according to the method of Yang et al. [74]. Briefly, 100 mL of test solution and 10,000 rotifers were added into a beaker, and the concentration of 0.1-μm and 1-μm MPs was 2000 μg/L. The beaker with sterile seawater was used as the control. After 24 h of exposure, all rotifers were collected and washed with sterile seawater. Then, 80% ethanol (*v*/*v*) was added to narcotize the rotifers. Samples were immersed in Bouin’s solution, and, after 6 h, 70% ethanol was used to wash the rotifers until there was no yellow color. Subsequently, rotifers were subjected to hematoxylin staining, dehydration with graded alcohol, and paraffin embedding. After staining with eosin, the sections were photographed under inverted microscopy. 

#### 4.3.3. Life-Cycle Test

The concentrations of MPs for 0.1-μm and 1-μm particles were set as 20, 200, and 2000 μg/L, while sterilized seawater without MPs was used as the control group. For each group, 1 mL of sterile seawater containing a series of MPs concentrations was injected into one well of a 24-well plate. Ten robust rotifers larvae (<2 h) were selected and added according to experimental systems established. The culture conditions were similar to those described for the hatching process. There were three replicates for each treatment. The number of surviving adults and larvae was recorded every 12 h under a stereomicroscope (XTL-400, Guiguang Instrument Co. LTD, Guilin, China), and the dead females and larvae were removed until all 10 females died. The rotifers were fed with *Chlorella* sp. (1.0 × 10^6^ cells/mL). To prevent microplastics and algae from aggregating during the experiment, the sample was shaken every 12 h to homogenize the solution, and 50% of the solution was changed every 24 h. The average lifespan and the total number of offspring were calculated [75,76,77].

### 4.4. Coexposure Test of 1 μm MPs and TCEP

#### 4.4.1. Population Growth

Two batches of experiments were carried out. It was reported that the concentration of TCEP in the River Ruhr, Germany, was up to 0.13 mg/L [78]; thus, 100 μg/L was used. In one experiment, three treatments, 1-μm MPs (20 μg/L), TCEP (100 μg/L), and 1-μm MPs + TCEP, were set, and the group with sterile seawater was considered as the control. In the other experiment, the combined effect of MPs and TCEP at high concentrations was also assessed to obtain more direct evidence. The LC_50_ value of TCEP was 672 mg/L, and the details and results of the acute toxicity test are shown in Appendix A when 65 mg/L (approximately 1/10 LC_50_) was used. Three treatments, 1-μm MPs (2000 μg/L), TCEP (65 mg/L), and 1-μm MPs + TCEP, were evaluated in this section.

During the experiment, four rotifer larvae (<2 h) per well were selected and inoculated into 12-well culture plates that contained 4 mL of test solution. During the experiment, half of the culture solution was renewed daily and fed with 1.0 × 10^6^ cells/mL *Chlorella* bait. The surviving individuals (including newborn larvae) were counted and observed daily under a stereomicroscope, and dead individuals were removed until the population reached a plateau stage of growth. After fitting with a logistic equation, the environmental capacity (K) and the time to reach the inflection point (Tp) were obtained (SI). 

#### 4.4.2. Oxidative Stress

The level of ROS in rotifers was measured by fluorometric analysis according to Zhang et al. [79]. A total of 120 robust adult rotifers were selected and randomly divided into four groups with three replicates. The experiment was conducted in a 24-well plate with 1 mL of sterile seawater containing the test concentrations of 1-μm MPs (2000 μg/L), TCEP (65 mg/L), and 1-μm MPs + TCEP. After 24 h of exposure, the rotifers were washed with sterile seawater and transferred to a new 24-well plate by adding the probe DCFH-DA at a final concentration of 20 μmol/L. They were incubated at 25 °C without light for 40 min and washed three times with PBS buffer solution. Then, the samples were fixed with 4% glutaraldehyde and photographed with a fluorescence microscope. The fluorescence intensity was measured by Image-Pro Plus 6.0.

The malondialdehyde (MDA) content was determined with an MDA detection kit (Nanjing Jiancheng Bioengineering Institute, Nanjing, China). The reaction between MDA and thiobarbituric acid (TBA) could produce the red adduct MDA–TBA, which was excited at 535 nm to produce maximum emission at 553 nm for fluorescence detection. Approximately 10,000 rotifers were added into a beaker with 100 mL sterile seawater. There were one control and three treatments: 1-μm MPs, TCEP, and 1-μm MPs + TCEP. Each group was evaluated in triplicate. After exposure for 24 h, all rotifers were collected and washed three times with PBS buffer. Following the procedures of the manufacturer, the samples were homogenized in an ice bath with an ultrasonic cell disruption system (UP-400s), and the best working parameters were set as 5 s operation and 10 s intervals, five times in total. Subsequently, the samples were centrifuged at 4 °C at 2500 rpm for 10 min, and the supernatant was kept for use as crude enzyme. The MDA content of each sample was normalized to the protein level (μmol/mg protein). The protein content was determined by a BCA Protein Kit (Nanjing Jiancheng Bioengineering Institute, Nanjing, China).

#### 4.4.3. Transcriptomic Analysis

##### RNA Extraction and RNA-seq

After 24 h of exposure to 1-μm MPs, TCEP, and 1-μm MPs + TCEP, approximately 10,000 rotifers in each group were collected in 1.5 mL RNA-free centrifuge tubes, quickly frozen in liquid nitrogen, and then stored at −80 °C until use. Total RNA was isolated using the mirVana™ miRNA Isolation Kit (Ambien) and digested with DNase I. RNA integrity was assessed using an Agilent 2100 Bioanalyzer 9 (Agilent Technologies, Santa Clara, CA, USA). After quality control, a cDNA library was constructed by PCR amplification. Sequencing was performed using an Illumina HiSeqTM 2500 sequencer, which was completed by OE Biotech Co., Ltd. (Shanghai, China).

##### Analysis of Differentially Expressed Unigenes (DEGs)

After transcriptome sequencing, Trimmomatic software was used to filter raw data for high-quality clean reads, and then the clean reads were de novo assembled into transcripts using Trinity with a paired-end method [80]. According to the similarity and length of a sequence, the longest one was chosen as a unigene for subsequent analysis. Gene expression levels were normalized by the FPKM method, and genes with *p* < 0.05 and |log_2_ (fold change)| > 1 were considered differentially expressed using the edgeR tool. Gene ontology (GO) functional enrichment and KEGG metabolic pathway analyses for DEGs were conducted.

##### Validation of Quantitative Real-Time PCR (qRT-PCR)

Six DEGs were randomly selected to confirm the reliability of transcriptome data by qRT-PCR. All primer sequences for qRT-PCR are listed in Appendix A. Reactions were performed using the PerfectStartTM Green qPCR SuperMix kit on a LightCycler^®^ 480 II fluorescent qPCR instrument (Roche, Basel, Switzerland). The total volume of the amplification reaction system was 10 μL, including 5 μL of 2× PerfectStartTM Green qPCR SuperMix, 0.2 μL of 10 μM forward primer, 0.2 μL of 10 μM reverse primer, 1 μL of cDNA, and 3.6 μL of nuclease-free H_2_O. qRT-PCR was performed at 94 °C for 30 s, 94 °C for 5 s, and 60 °C for 30 s, followed by 45 cycles. At the end of the cycles, the product specificity was detected using the melting curve; the temperature was slowly raised from 60 °C to 97 °C, and the fluorescence signal was collected five times per °C. Each experiment was conducted in triplicate, and the expression level of DEGs was normalized using the 2^−ΔΔCT^ method.

#### 4.4.4. Activities of MXR Key Components

##### P-gp and MRP Activity

The activity of P-gp and MRP was measured according to Jeong et al. [7] with some modifications. Rhodamine B and calcein AM are widely used as fluorescent substrates of P-gp and MRP, respectively. When these fluorescent substrates enter the organism, P-gp/MRP are activated to expel these substrates; thus, the retention values of fluorescence in the organism are inversely proportional to the efflux activity of P-gp/MRP, and vice versa. After 24 h of exposure to 1-μm MPs, TCEP, and 1-μm MPs + TCEP, approximately 10,000 rotifers in each group were collected and incubated with rhodamine B (Sigma-Aldrich, Saint Louis, MO, USA) at 0.5 μmol/L or calcein AM (Sigma-Aldrich, Saint Louis, MO, USA) at 1 μmol/L for 2 h. After incubation, the rotifers were washed three times and homogenized in PBS. The samples were centrifuged at 10,000× *g* for 10 min, and the supernatant was kept, where the rhodamine B/calcein AM level represented the retention portion. Then, 200 μL of lipid was measured using a multimode reader (PerkinElmer, New York, NY, USA) with an excitation wavelength of 535 nm and an emission length of 590 nm. According to a calibration curve of rhodamine B/calcein AM, the fluorescence intensities of stranded rhodamine B/calcein AM (F.a.v./mg protein) were normalized to the protein content.

##### GST Activity

The activities of GST were determined with a detection kit (Beijing Box Biotech Co., Ltd. Beijing, China). Reduced glutathione (GSH) can combine with 1-chlom-2,4-dinitrobenzene (CDNB) in the presence of GST to generate GS-DNB with a characteristic absorption peak at 340 nm. After 24 h of exposure, approximately 10,000 rotifers were collected and crushed for crude enzyme according to the manufacturer’s instructions. After the reaction, the absorbance value of the liquid was measured at 340 nm by multimode readers in a 96-well UV plate. The unit of GST activity (U/mg protein) was defined as 1 µmol of CDNB bound to GSH catalyzed per mg protein per min.

### 4.5. Statistical Analysis

All data are presented as the mean ± standard deviation (SD) of three replicates, and SPSS 25 (SPSS Inc., Chicago, IL, USA) software was used in the statistical analysis of the data. The differences among experimental groups were examined by one-way analysis of variance (ANOVA) and Tukey’s multiple comparisons; the level of *p* < 0.05 was considered significant. All graphs were generated using Origin 2018 (Origin Lab. Corporation, Northampton, MA, USA).

## 5. Conclusions

In summary, 0.1-μm and 1-μm MPs at concentrations ranging from 20 to 2000 μg/L did not exert obvious effects on the survival, reproduction, and population growth of *B. plicatilis*. The unaffected microstructure of the digestive system and increased transcriptional expression of five *GSTs* might have accounted for this limited effect of MPs on *B. plicatilis*. TCEP inhibited population growth and induced oxidative stress in the rotifers, while 1-μm MPs + TCEP did not affect them. The genes involved in metabolic detoxification (e.g., *GST S4* and *Abcb1*), stress response (*Hsp70* and *Hsp90*), and damage repair (*MSH5*) were more significantly upregulated by 1-μm MP + TCEP stress than by TCEP stress. Co-exposure also notably activated the efflux ability of P-gp. Our results issue a challenge to the traditional view that MPs increase the toxicity of pollutants. Furthermore, this study can supply a potential molecular basis for the elucidation of the underlying mechanism of the reversal of TCEP toxicity in the presence of MPs.

## Figures and Tables

**Figure 1 ijms-23-04934-f001:**
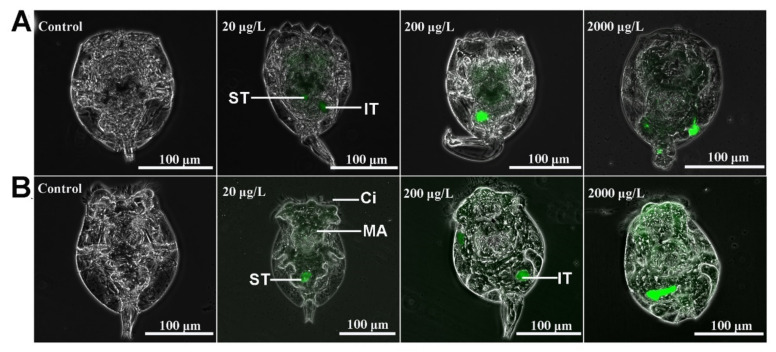
Fluorescently labeled MPs with sizes of 0.1 μm (**A**) and 1 μm (**B**) were observed in rotifers. Ci, cilia; IT, intestine; MA, mastax; ST, stomach.

**Figure 2 ijms-23-04934-f002:**
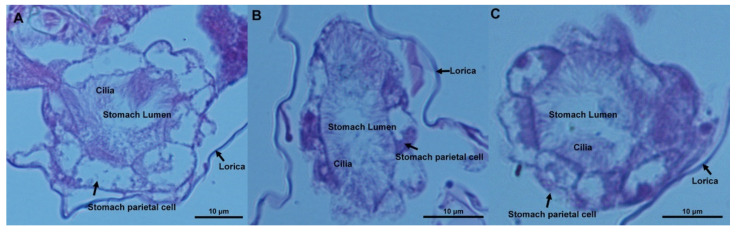
Microstructural images of rotifer stomach in the control group (**A**), 0.1-μm MP group (**B**), and 1-μm MP group (**C**). The hyperchromatic areas represent the nuclei of the stomach parietal cells of *Brachionus* because they were syncytia.

**Figure 3 ijms-23-04934-f003:**
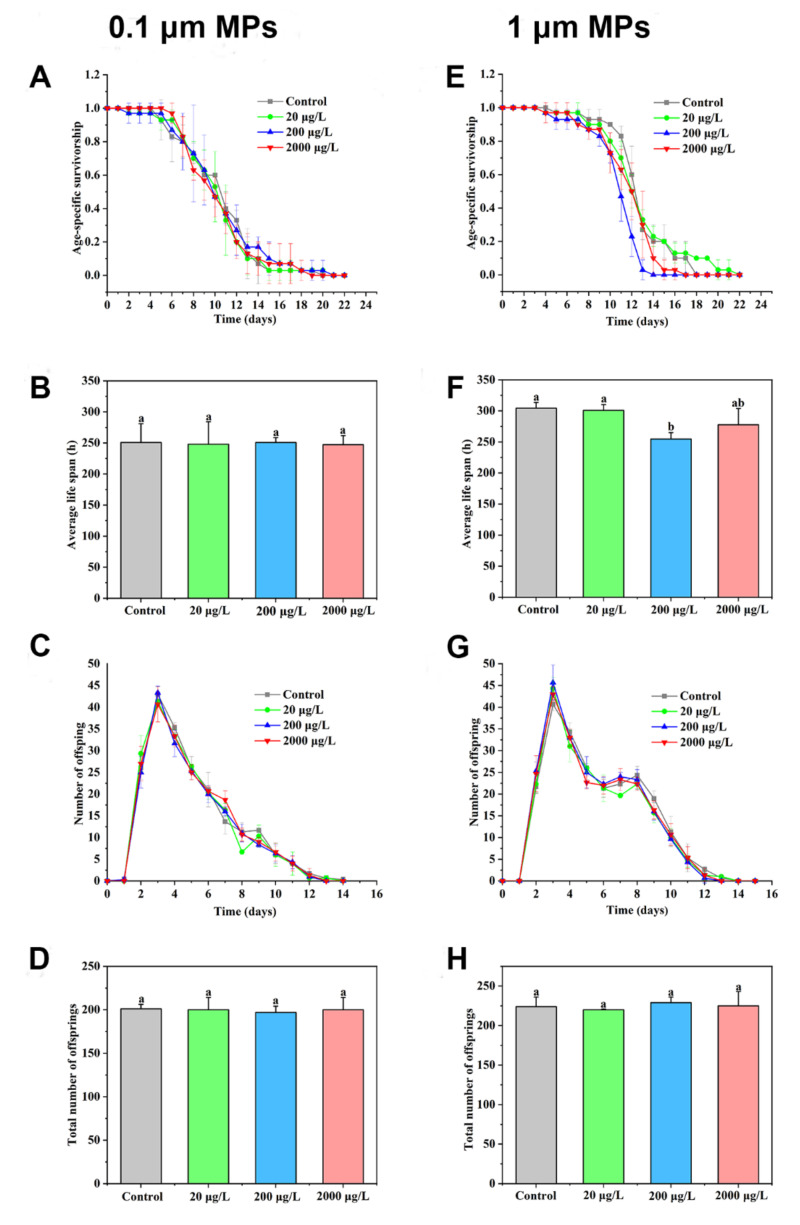
Effect of different concentrations of 0.1-μm MPs (**A**–**D**) and 1-μm MPs (**E**–**H**) on the survival and reproduction of rotifers (n = 30): (**A**,**E**) age-specific survivorship; (**B**,**F**) average lifespan; (**C**,**G**) daily number of offspring; (**D**,**H**) total number of offspring. Data are expressed as the mean ± SD of three parallel experiments. Different lowercase letters represent significant differences between experimental groups (*p* < 0.05).

**Figure 4 ijms-23-04934-f004:**
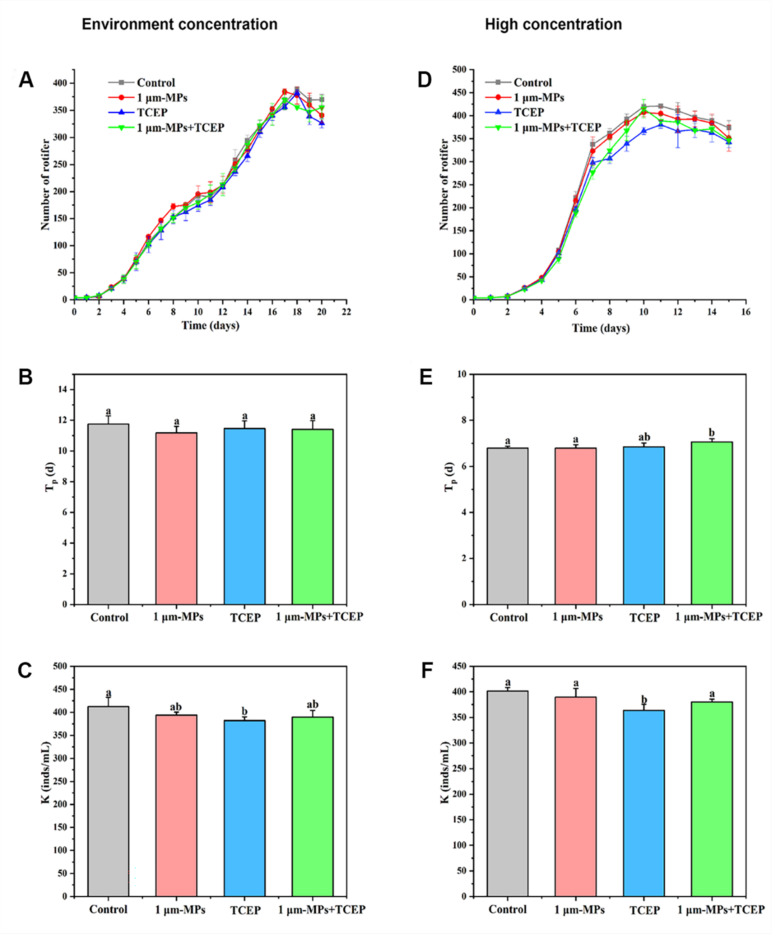
Population growth of *B. plicatilis* and its related parameters in response to 1-μm MP and TCEP exposure at environment-related concentrations (MPs, 20 μg/L; TCEP, 100 μg/L) and high concentration (MPs, 2000 μg/L; TCEP, 65 mg/L) (n = 12): (**A**,**D**) population growth curve; (**B**,**E**) inflection point of growth curve, Tp value; (**C**,**F**) environmental carrying capacity, K value. Data are expressed as the mean ± SD of three parallel experiments. Different lowercase letters represent significant differences between experimental groups (*p* < 0.05).

**Figure 5 ijms-23-04934-f005:**
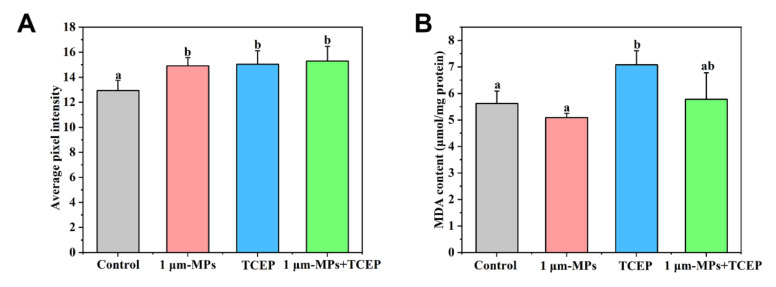
Effect of 1 μm MPs + TCEP at high concentration (MPs, 2000 μg/L; TCEP, 65 mg/L) on oxidative status of *B. plicatilis* (n = 30,000): (**A**) ROS level; (**B**) MDA content. Data are expressed as the mean ± SD of three parallel experiments. Different lowercase letters represent significant differences between experimental groups (*p* < 0.05).

**Figure 6 ijms-23-04934-f006:**
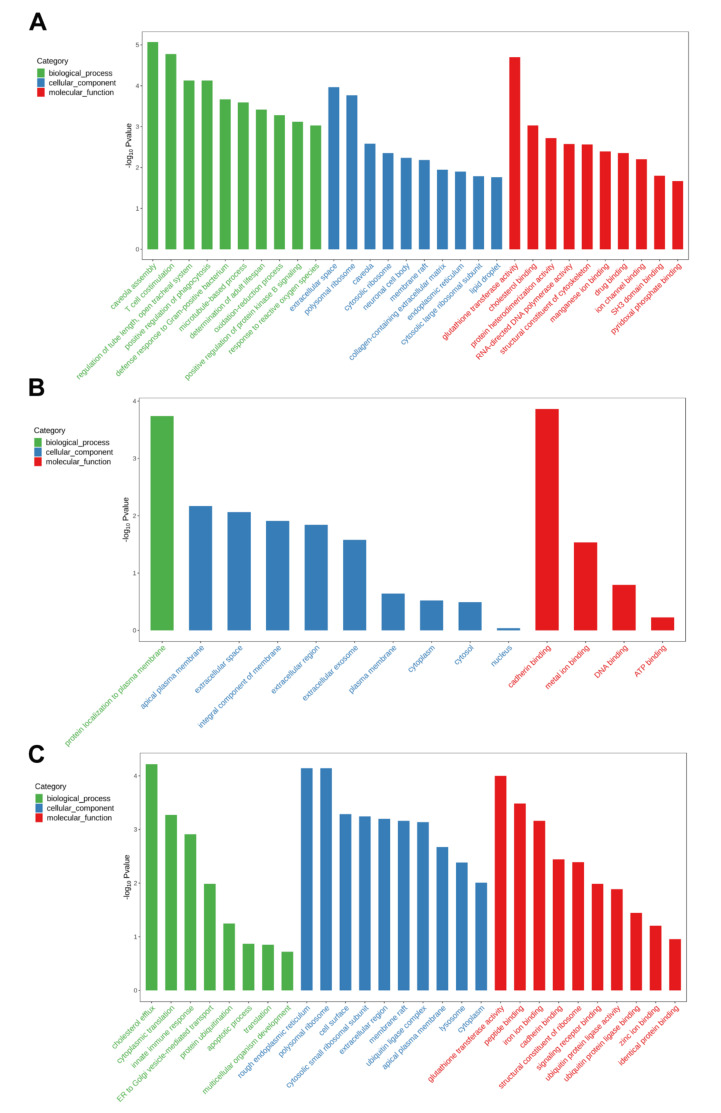
The top 30 GO terms: (**A**) 1-μm MPs; (**B**) TCEP; (**C**) 1-μm MPs + TCEP. The horizontal axis represents the GO entry name, and the vertical axis represents the −log (*p*-value).

**Figure 7 ijms-23-04934-f007:**
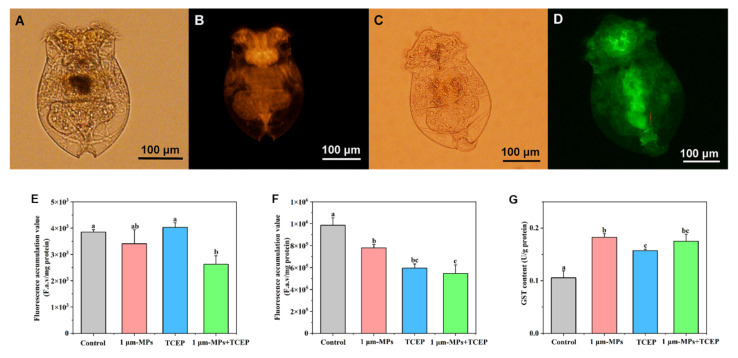
Effect of 1-μm MPs + TCEP at high concentration on the activities of MXR key components of *B. plicatilis* (n = 30,000). Images of rotifers after staining of rhodamine B (**A**,**B**) and calcein AM (**C**,**D**): (**A,C**) rotifers under light microscope; (**B**,**D**) rotifers under fluorescence microscope; (**E**) retention amounts of rhodamine B; (**F**) retention amounts of calcein AM; (**G**) GST activities. Data are expressed as the mean ± SD of three parallel experiments. Different lowercase letters represent significant differences between experimental groups (*p* < 0.05).

**Figure 8 ijms-23-04934-f008:**
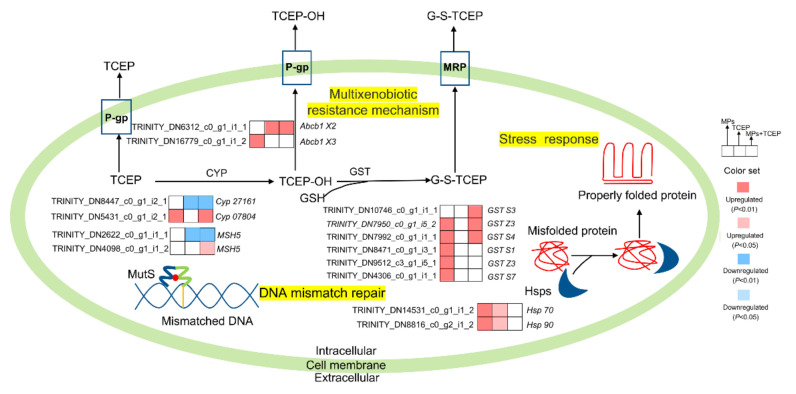
Overview of transcriptional changes in gene that matched to *Brachionus* when rotifers were co-exposed to MPs and TCEP. Each index comprises the changes in gene transcriptions in different exposure groups (MPs, TCEP, and MPs + TCEP) relative to the control group. The red or blue color scale indicates gene transcripts shown to be up- or downregulated, respectively.

## Data Availability

The data presented in this study are available in the article.

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
