# Peer review of "Alleviation of Tris(2-chloroethyl) Phosphate Toxicity on the Marine Rotifer Brachionus plicatilis by Polystyrene Microplastics: Features and Molecular Evidence"

_ijms, 2022, doi:10.3390/ijms23094934_

Round 1
Reviewer 1 Report
Dear authors,
Title: Alleviation of tris(2-chloroethyl) phosphate toxicity on the marine rotifer Brachionus plicatilis by polystyrene microplastics: features and molecular evidence
This is an interesting study about a very current topic microplastics pollution. I think there are some oversights in methods and results. I do not understand how many individuals were used for each analysis, but 10 (Life-Cycle test) or 4 (Population Growth) maybe is not enough. You must specify the number of samples for each analysis. You can do this in Results, in figures and in text also.
The microstructural photos are not notable, there are some differences in size and structures of stomach parietal cells and lumen.
There are no results for Anova and Tukey’s multiple comparisons tests presented in the manuscript.
There are also some specific observations:
Lines 111-118: Can you give some values? How much MPs did they accumulate? Can you quantity the MPs accumulation? Did you used any test for comparison between the study groups? Maybe Anova and Tukey’s multiple comparisons tests?
Line 119: The figure should have self-explaining captions. What represent the green colour? You should add also in the figure caption.
Line 130: These 3 stomachs look quite different. Is there any difference in size? Stomach parietal cells looks quite different in B and C comparative with A.
Lines 136 – 137: Can you present the results of the Anova and Tukey’s multiple comparisons tests? If not sufficient just to say ” No difference occurred between the control and 0.1-μm MPs treat-136 ments, even though the concentration of MPs was up to 2,000 μg/L”
Lines 147-151: Can you also add the number of individuals tested in each group?
Line 175: What means environment-related concentration and high concentration?
Lines 174 – 178: Can you also add the number of individuals tested in each group?
Line 179: Can you also add the number of individuals tested in each group?
Line 215: The figure should have self-explaining captions.
Lines 240 – 245: Can you also add the number of individuals tested in each group?
Lines 253 – 255: Is better to avoid figure or table citation in Discussion part.
Lines 378 – 385 How many individuals were checked for digestive system?
Line 390: Are ten individuals enough to make this analysis?
Line 409: Are four rotifer larvae enough?
Best regards,
Reviewer 2 Report
Due to the increasing pollution of the environment with microplastics (MPs), there is growing interest in assessing the influence of the presence of microplastics on the toxicity of various toxic substances. In the peer-reviewed study assessed the impact of mixture of organophosphorus ester (TCEP) and two size (0.1 um and 1 um) microplastics on the marine rotifer Brachionus plicatilis. The advantage of the work is the use of a wide range of test responses, from mortality and inhibition of population growth rate, through the assessment of biochemical parameters, to transcriptomics.
General comment. The concentration of MPs was expressed in ug/l. However, the effect depends on the number of particles. For both types of particles, the conversion of concentration in mg/l into concentration expressed as number of particles in ml should be given.
Paragraph 2.1. It cannot be written that bioaccumulation took place, only that the organisms took up MPs. Bioaccumulation occurs when the concentration of a xenobiotic increases over time and it cannot be excreted. Have you studied if and when the rotifers excreted the collected MPs?
Figure 3. Why are the survival and reproduction curves different for the control samples at 0.1 um and 1 um MPs?
Paragraph 2.6. How was it supposed that MP was inside the cells and activated the efflux pumps?
4.4.2. Line 423. Why were the rotifers incubated at 37 C?
Line 431. What was the volume of the sample? What was the concentration of the organisms?
Line 514. MPs has not been shown to absorb the toxic substance.
Suplementary information.
- The information on the MPs concentration expressed in particles/ml should be given. Was the suspension homogeneous? Did it not aggregate, especially in the presence of algae cells?
Round 2
Reviewer 1 Report
Dear Authors,
Thank you for your responses and also for your correction in the second form of the manuscript: “Alleviation of tris(2-chloroethyl) phosphate toxicity on the marine rotifer Brachionus plicatilis by polystyrene microplastics: features and molecular evidence” (No. ijms-1677437).
I have few more recommendations for the present form of the manuscript:
- The Introduction section could be improved. In the R1-1 table you have 3 studies on Brachionus plicatilis and some on Brachionus koreanus which are not cited in your manuscript.
- Please add in the Materials and Methods section that your experimental system is based on the protocol of American Society for Testing and Materials (ASTM): Standard Guide for Acute Toxicity Test with Rotifer Brachionus (Designation: E1440-91).
- The “Table R1-1Summary of experimental system in ecotoxicological studies targeting at rotifers” is a nice information which maybe you can integrate in the Supplementary Materials, including here also the present study.
- Please add the explanation for Point 2 in the manuscript or you can add a short explanation in the Figure caption.
- For One-way ANOVA please add also the “F” and “df” values in the brackets, near the “p” values.
